# Disease coverage of human genome-wide association studies and pharmaceutical research and development
María Gordillo-Marañón [1] ✉, Amand F. Schmidt [1,2,3], Alasdair Warwick[1], Chris Tomlinson [4], Cai Ytsma [4], Jorgen Engmann [1], Ana Torralbo[4], Rory Maclean [4], Reecha Sofat [5,6], Claudia Langenberg [7,8,9], Anoop D. Shah [4,10], Spiros Denaxas[4,10,11], Munir Pirmohamed [12], Harry Hemingway [4,6,10], Aroon D. Hingorani [1,3,13] & Chris Finan [1,3,13]

## Abstract

**Background** Despite the growing interest in the use of human genomic data for drug target identification and validation, the extent to which the spectrum of human disease has been addressed by genome-wide association studies (GWAS), or by drug development, and the degree to which these efforts overlap remain unclear.
**Methods** In this study we harmonize and integrate different data sources to create a sample space of all the human drug targets and diseases and identify points of convergence or divergence of GWAS and drug development efforts.
**Results** We show that only 612 of 11,158 diseases listed in Human Disease Ontology have an approved drug treatment in at least one region of the world. Of the 1414 diseases that are the subject of preclinical or clinical phase drug development, only 666 have been investigated in GWAS. Conversely, of the 1914 human diseases that have been the subject of GWAS, 1121 have yet to be investigated in drug development.
**Conclusions** We produce target-disease indication lists to help the pharmaceutical industry to prioritize future drug development efforts based on genetic evidence, academia to prioritize future GWAS for diseases without effective treatments, and both sectors to harness genetic evidence to expand the indications for licensed drugs or to identify repurposing opportunities for clinical candidates that failed in their originally intended indication.

## Plain language summary

The pharma industry has shown growing interest in the use of human genomic data to support drug development and reduce the risk of clinical-stage failure. We investigate the extent to which human diseases have been the subject of genetic studies, of pharmaceutical research and development, or both. We show that only a small proportion of all human diseases have an approved drug treatment and that less than half of all the diseases that are the subject of preclinical or clinical phase drug development have been investigated in genetic studies. In addition, approximately two-thirds of the diseases covered in genetic studies have yet to be investigated in drug development. These findings could help prioritize drug development efforts or genetic studies for diseases without effective treatments.

Pre-clinical, cell and animal model-based approaches for drug target identification and validation have been poorly predictive of human efficacy, contributing to the high failure rate in clinical phase drug development[1–3] due to lack of benefit or unanticipated adverse effects[4,5].

Human genetics may help improve drug development efficiency by (i) helping to map drug targets to diseases more accurately and systematically through genome-wide association studies (GWAS) (target identification); and (ii) using DNA sequence variants in a gene encoding a drug target, that

---

[1]Institute of Cardiovascular Science, Faculty of Population Health, University College London, London, United Kingdom. [2]Department of Cardiology, Amsterdam Cardiovascular Sciences, Amsterdam University Medical Centres, University of Amsterdam, Amsterdam, the Netherlands. [3]UCL British Heart Foundation Research Accelerator, London, United Kingdom. [4]Institute of Health Informatics, Faculty of Population Health, University College London, London, United Kingdom. [5]Department of Pharmacology and Therapeutics, University of Liverpool, Liverpool, United Kingdom. [6]Health Data Research, London, United Kingdom. [7]Precision Healthcare University Research Institute, Queen Mary University of London, London, United Kingdom. [8]Computational Medicine, Berlin Institute of Health at Charité Universitätsmedizin, Berlin, Germany. [9]MRC Epidemiology Unit, University of Cambridge, Cambridge, United Kingdom. [10]NIHR Biomedical Research Centre at University College London Hospitals, London, United Kingdom. [11]British Heart Foundation Data Science Centre, London, United Kingdom. [12]Department of Pharmacology and Therapeutics, Centre for Drug Safety Science, University of Liverpool, Liverpool, United Kingdom. [13]These authors contributed equally: Aroon D. Hingorani, Chris Finan. ✉e-mail: maria.maranon.16@ucl.ac.uk

influence its expression or function, to anticipate the full range of beneficial and harmful mechanism-based effects of a drug acting on the encoded protein (target validation), using drug target Mendelian randomisation[6–10]. Several lines of empirical evidence support this concept: (a) Many GWAS have rediscovered established drug targets for the corresponding diseases[11–13]; (b) Target–disease pairings with genetic support are enriched among successful drug development programmes[14–18]; (c) Comparative studies have shown that the effect of licensed drugs on biomarkers and disease endpoints coincide with the observed associations of variants in the genes encoding the corresponding target[19–21]; and (d) Several drugs have now been successfully developed or repurposed on the basis of human genetic evidence (e.g., maraviroc for treatment of HIV infection[22,23]; PCSK9 inhibitors for hypercholesterolaemia and coronary disease prevention[20,24] and tocilizumab for treatment of pro-inflammatory adverse outcomes of SARS-CoV-2 infection[25,26]). For these reasons, the pharmaceutical industry has shown a growing interest in the use of human genomic data to help prioritise drug development programmes and reduce the risk of clinical-stage failure.

Pharma partnerships have provided substantial investment for sequencing, genotyping or molecular phenotyping of large national bio-banks, which are linked to routinely collected primary and secondary care electronic health records (e.g., in the UK[27] and Finland[28]). Some have engaged in partnerships with healthcare providers (e.g., Regeneron with Geisinger Healthcare in the US). Others have partnered with consumer genetic testing companies (e.g., GSK with 23andMe[29]). Several pharmaceutical companies have also invested in Open Targets, a partnership with the European Bioinformatics Institute and the Welcome Trust Sanger Institute that seeks to harness summary-level genetic association data from GWAS to inform therapeutic hypotheses[13].

However, until recently, genetic studies of human diseases and pharmaceutical research and development have largely proceeded independently. Thus, the extent to which the causes of human disease have been addressed by genetic analyses, or by drug development, and the degree to which these efforts overlap, has not been investigated systematically. Filling this gap in knowledge could have several applications. First, a survey of this type would help understand where future drug development programmes could be directed if they are seeking to exploit existing genetic evidence on therapeutic targets. Second, it could also help prioritise new, large-scale GWAS or sequencing studies to help stimulate drug development for diseases currently without effective treatments. Finally, it could help quantify and inform opportunities to expand the indications for licensed drugs or identify repurposing opportunities for the many safe drugs that failed in clinical trials because of lack of efficacy in the originally intended indication. To address this gap in knowledge, we connected disparate sources of data to evaluate disease coverage and the overlap of GWAS and pharmaceutical research and development. We find that only 5% of the diseases have an approved drug treatment in at least one region of the world and that less than half of the diseases that pharmaceutical companies are currently attempting to develop drugs for have been investigated by GWAS. In addition, approximately two-thirds of the diseases investigated using genetic studies have yet to be investigated in drug development. We produce drug target-disease indication lists to help prioritize drug development efforts or genetic studies for diseases without effective treatments.

## Methods
### Human diseases
To estimate the total number of human diseases, we retrieved information from widely used disease classification systems and ontologies (Table 1). As of 29 September 2022, the Human Disease Ontology (DO)[30] had 11,158 disease terms. Since the number of terms in the DO is curated and updated regularly, the rationale described in previous studies[31] was followed, and a figure of 11,158 was proposed as a reasonable estimate of the number of common human diseases with genetic susceptibility.

### Drug and target data
Compound, target and drug indication data (where relevant) were extracted from ChEMBL version 31 (v31). ChEMBL includes compounds under both preclinical and clinical development. Information in ChEMBL is itself based on several resources, including United States Adopted Name (USAN) applications, ClinicalTrials.gov; the FDA Orange Book database, the British National Formulary, and the ATC classification for compounds with a license. ChEMBL was selected as the data source for compound, target and indication information for the following reasons: (i) it is publicly available, (ii) it is manually curated, (iii) it contains information on drugs that have been approved for the treatment of a specific disease/diagnosis (an indication) within any region of the world. Pharmaprojects[32] or Patsnap[33] are private resources and, thus, were not considered for this analysis. ChEMBL was selected over DrugBank as the latter only contains information on FDA-approved drugs[34]. Since proteins are the major category of drug targets, drug targets were mapped to the corresponding UniProt identifiers, and thence to gene identifiers in Ensembl version 95 (GRCh37) through the updated druggable genome[11]. Compounds flagged as withdrawn or not intended for human use were excluded from the analysis. Diseases with an approved treatment and/or a treatment under clinical or preclinical development were sourced from ChEMBL v31[35], which provided standardised indication terms based on MeSH. MeSH terms in ChEMBL v31 were mapped to Unified Medical Language System (UMLS)[36] concepts using the MRCONSO table in the UMLS2022AA.

### GWAS data
The collection of traits studied by GWAS was obtained from the GWAS Catalog v1.0.3, which represented the most complete source of publicly available GWAS[37]. The dataset was expanded with GWAS summary statistics from UK Biobank (Neale data, GWAS Round 2, Results shared 1st August 2018)[38] to include diseases that were not covered by established case/control cohorts.

The GWAS Catalog v1.0.3 comprised 5580 unique traits for 6041 PubMed publications, including diseases, biomarkers, molecular measurements, drug responses and anthropometric measurements. Most trait descriptions (reported traits) in the GWAS Catalog are mapped to terms from the Experimental Factor Ontology (EFO), however, the mapping also includes other ontologies: 'GO', 'MONDO', 'HP', 'Orphanet', 'PATO', 'NCBITaxon', 'MP', 'NCIT', 'UBERON', 'OBA', 'HANCESTRO'. To filter human diseases from the 5580 traits in the GWAS Catalog, terms were mapped to UMLS concepts using several complementary approaches. First, 1440 traits were mapped to 1440 UMLS concepts using direct string matching to MeSH terms and the MRCONSO table in the UMLS version 2022AA (UMLS2022AA). Next, 994 traits were mapped to 1357 UMLS concepts using Batch MetaMap, which used UMLS version 2020AB[38,39]. For traits mapping to multiple UMLS concepts, those resulting in a direct match were selected ($n = 620$), followed by those with a UMLS semantic type of disease or syndrome[40] ($n_{traits} = 121$, $n_{UMLS concepts} = 133$), neoplastic process ($n_{traits} = 48$, $n_{UMLS concepts} = 62$), Mental or Behavioral Dysfunction ($n_{traits} = 21$, $n_{UMLS concepts} = 24$), Congenital Abnormality ($n_{traits} = 2$, $n_{UMLS concepts} = 2$), Sign or Symptom ($n_{traits} = 9$, $n_{UMLS concepts} = 10$), Finding ($n_{traits} = 27$, $n_{UMLS concepts} = 30$), Laboratory Procedure ($n_{traits} = 77$, $n_{UMLS concepts} = 75$), Injury or Poisoning ($n_{traits} = 5$, $n_{UMLS concepts} = 4$), Individual behaviour ($n_{traits} = 4$, $n_{UMLS concepts} = 6$), diagnostic procedure ($n_{traits} = 9$, $n_{UMLS concepts} = 9$). The remaining 44 traits were mapped to 71 UMLS concepts. Therefore, the final mapping of the 994 traits resulted in 1050 UMLS concepts. Then, 146 traits were mapped to 163 UMLS concepts using the UMLS mappings in DisGeNET[41], 136 traits were mapped to 355 UMLS concepts using cross-mapping between ontologies in DisGeNET[41], and 942 were mapped to 833 UMLS concepts using the UMLS Metathesaurus[42]. The remaining 1922 traits were mapped to UMLS concepts using the MeSH terms associated with the PubMed publications. Because multiple MeSH terms can be associated with the study and do not represent the trait, to identify the MeSH term that corresponded to the

**Table 1 | The number of terms within common disease classification systems and ontologies as of 29 November 2022**

| Coding scheme | Type | Number of terms | Coverage |
|---|---|---|---|
| ICD-10[a] | Disease classification | 12,318 | Clinical disease classification is mainly according to appearance rather than cause |
| Human Disease Ontology (DO)[b] | Ontology | 11,158 | Biomedical resource of standardised disease concepts organised by disease aetiology |
| Human Phenotype Ontology (HPO)[a] | Ontology | 16,601 | Phenotypic abnormalities and clinical observations |
| Experimental Factor Ontology (EFO)[c] | Ontology | 40,133 | Experimental variables from the cellular to disease level in the European Bioinformatics Institute (EBI) databases |
| Medical Subject Headings (MeSH)[a] | Clinical terminology designed for indexing and cataloguing biomedical literature | 348,733 | Anatomy, organisms, diseases, chemicals and drugs, techniques and equipment, biological science, psychiatry and psychology, physical sciences, anthropology, education, sociology and social phenomena, technology and food and beverages, humanities, information science, health care |
| SNOMED CT[a] | Clinical terminology designed for recording clinical data in electronic health records (EHRs) | 498,686 | Clinical findings, symptoms, diagnoses, procedures, body structures, organisms and other etiologies, substances, pharmaceuticals, devices and specimens |
| Unified Medical Language System (UMLS)[a] | Biomedical vocabularies repository | 3,619,007 | Biomedical and health-related concepts by multiple source vocabularies |

[a]https://www.nlm.nih.gov/pubs/techbull/mj22/mj22_umls_2022aa_release.html.
[b]https://github.com/DiseaseOntology/HumanDiseaseOntology/tree/master/src/ontology.
[c]https://github.com/EBISPOT/efo/blob/master/ExFactor%20Ontology%20release%20notes.txt.

mapped trait, MeSH terms indicated as major terms were selected and manually curated against the mapped trait.

The 633 ICD-10 diagnoses in Neale data were automatically mapped to UMLS concepts using the UMLS2020AA MRCONSO table. In total, 983 unique diseases were identified and manually curated. The diseases were mapped to disease areas according to ICD10 chapters. Diseases classified in the chapters: "Animal diseases", "Findings, not elsewhere classified", and "Pregnancy, childbirth and the puerperium" were excluded, resulting in a total of 953 unique disease terms (Supplementary Data 5).

### Mapping between GWAS and drug data

To facilitate further mappings and estimate the coverage, overlap and divergence of human GWAS and diseases investigated in pharmaceutical research and development, disease UMLS concepts from GWAS data and indications UMLS concepts from ChEMBL were overlapped. The UMLS was selected as the anchoring coding system as it integrates several medical vocabularies to enable interoperability between data sources and facilitate the link between terms from different coding systems.

### Statistics and reproducibility

Analyses were conducted using Python v3.7.3 and Jupyter notebooks.

### Reporting summary

Further information on research design is available in the Nature Portfolio Reporting Summary linked to this article.

## Results

### Protein-coding genes and genes encoding drug targets ('drug-gable genome')

We first obtained estimates of the total number of protein-coding genes and, from this, we identified the subset of protein-coding genes considered to be most amenable to targeting by drugs, a subset of the protein-coding genome known as the 'druggable genome'[11]. At the time of the study, the total number of protein-coding genes in the human genome is estimated at 19,813 as annotated in Ensembl v.108; of which 4729 encode proteins estimated to be amenable to targeting by small molecule drugs or biotherapeutics[11]. Of all human genes encoding druggable targets, 755 (16%) are already the targets of approved drugs, 1218 (26%) are the targets of drugs in clinical development, 418 (9%) are targets of drugs in preclinical

development, with 3495 (73.9%) being currently 'undrugged' (Fig. 1, Supplementary Data 1). Data on drugs in preclinical development may be incomplete as information on many withdrawn targets is not publicly available.

### Human diseases evaluated in drug development and in GWAS

Producing plausible estimates for the total number of human diseases (the 'disease-ome') is challenging due to the hierarchical nature of biomedical vocabularies, duplications and descriptive terms beyond diagnoses present in clinical terminologies and disease classification systems. In 2019, a figure of 10,000 was proposed as a reasonable estimate of the number of common human diseases with genetic heritability[31]. Here, an updated figure of 11,158 diseases was used, which corresponded to the number of terms overall levels in the Disease Ontology (DO)[30], an open-source ontology of human diseases that is updated regularly, coordinated by the University of Maryland Institute for Genome Sciences. This is likely to be a slight overestimation given that parent terms are included, however, it circumvents the need to decide on the granularity of the disease term. We focused on 11,158 diseases in the DO as the denominator, as this set encompasses diseases more likely to be studied using a GWAS design. We could have alternatively used ICD-10 or Human Phenotype Ontology (HPO) to perform the calculation, which would not have impacted the findings, as they include a very similar number of diseases compared to DO.

ChEMBL is an open-access drug database that contains information on drugs that have been approved for the treatment of a specific disease/diagnosis (an indication) within a region of the world and clinical candidate drugs that are being or have been investigated for an indication. By sourcing data from the ChEMBL v31[35], we found that only 1549 unique diseases (estimated as 13.9% of the total number of human diseases in DO) have an approved drug and/or a drug under clinical or preclinical development. This comprises 612 diseases that are the indication of approved drugs (Supplementary Data 2), 1401 diseases that are or have been the indication of drugs in clinical development (Supplementary Data 3) and 210 diseases that are or have been indications for drugs in preclinical development (Supplementary Data 4), as shown in Fig. 2. Note that some diseases can appear in more than one category if several compounds are studied for the same indication and are in different stages of the drug development process.

Equally, estimating the proportion of diseases covered by GWAS is difficult because some diseases could have been studied through a validated

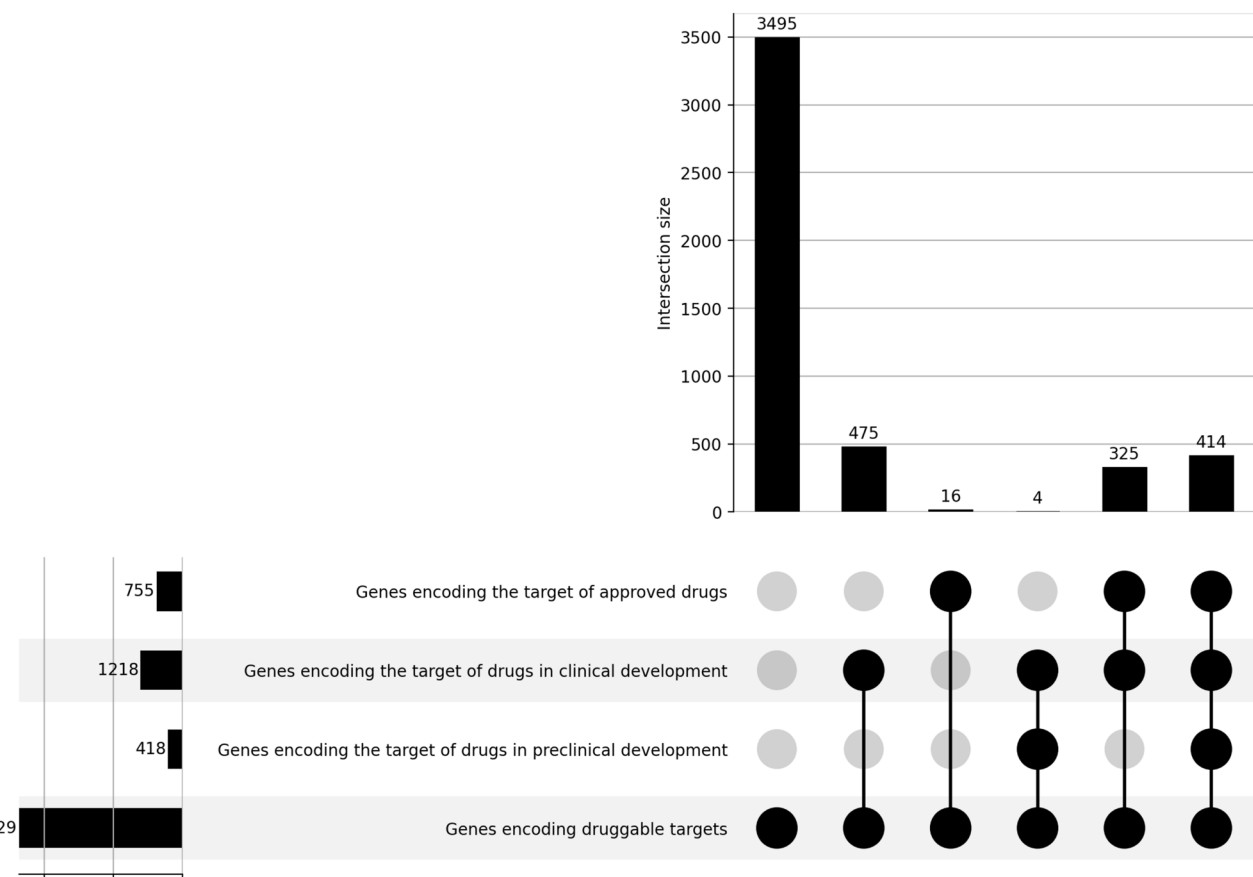

**Fig. 1 | Total count of genes encoding druggable targets, with subsets and intersection of genes encoding the targets of approved drugs, and drugs in clinical or preclinical development.** The figure shows the intersections of a set as a matrix, where each row represents a set, the bar charts on the left are the size of the set, and the bar charts on the top are the size of the intersection. Each column indicates a possible intersection, where the filled-in cells show which set is part of the intersection. The lines connecting the filled-in cells indicate the direction the plot should be interpreted.

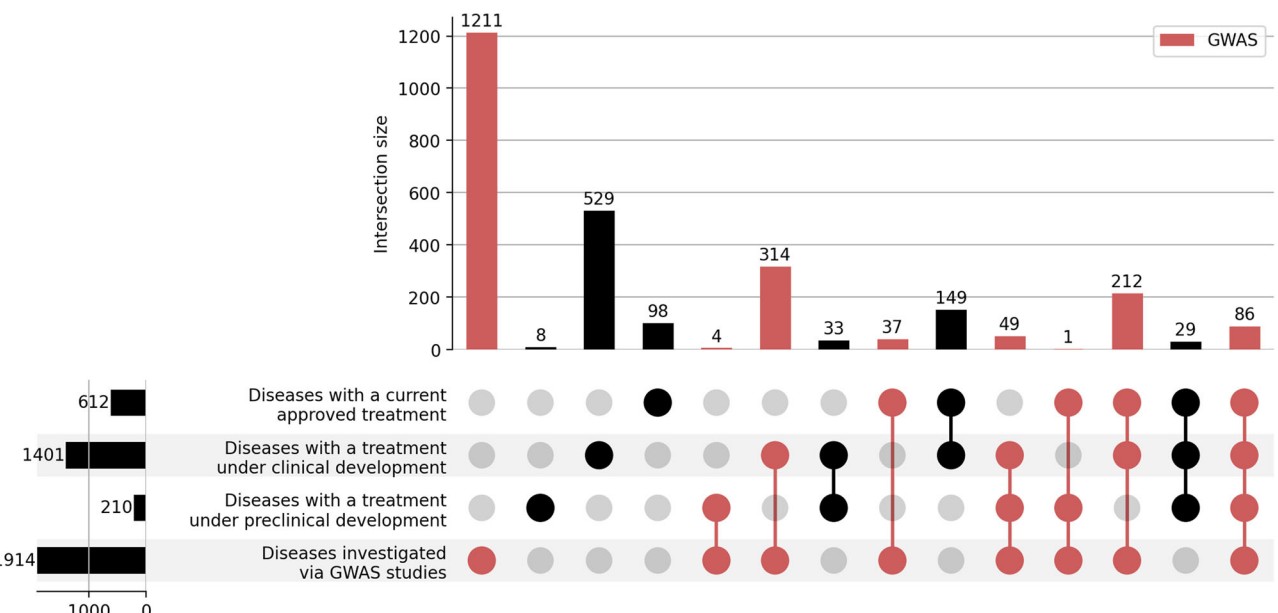

**Fig. 2 | Intersection between diseases with current approved treatment, with a treatment that is or has been under clinical development, with a treatment that is or has been under preclinical development, or investigated via GWAS.** The figure shows the intersections of a set as a matrix, where each row represents a set, the bar charts on the left the size of the set and the bar charts on the top the size of the intersection. Each column indicates a possible intersection, where the filled-in cells show which set is part of the intersection. The lines connecting the filled-in cells indicate the direction the plot should be read. Subsets, including diseases studied by GWAS, are indicated in red. Data sources: ChEMBL v31 (approved, clinical and preclinical development), GWAS Catalog v1.0.3 (GWAS studies) and UK Biobank through Neale data (GWAS studies).

**Fig. 3 | Illustration of the sample space and subsets of human proteins and diseases.** The complete sample set (**A**) is bounded by the total number of protein-coding genes (19,995) and the sum total of common, complex human diseases (11,158). The subset of all potentially druggable target–disease indication pairings is indicated by subset **B**, the drug target–disease indication pairings studied in clinical phase drug development by subset **C**, and the target-disease indication pairings of approved drugs by subset **D**. The vertical lines represent diseases studied by GWAS on the assumption that GWAS interrogate all genes in the human genome (subsets **E** and **F**). The presence of two GWAS subsets illustrates the point that only a subset of diseases studied in GWAS have also been the subject of drug development (**E**). See text for further explanation.

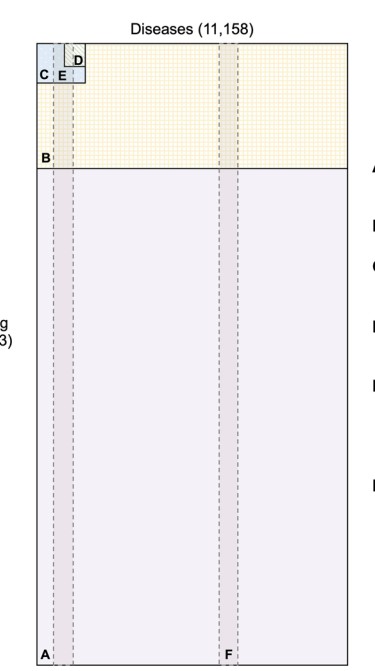

**A -** Full sample space of protein-coding gene - disease pairs

**B -** Genes encoding druggable targets

**C -** Drug target gene - disease pairs with a drug under clinical investigation

**D -** Drug target gene - disease pairs with an approved drug

**E -** Drug target gene - disease pairs studied by GWAS where the disease has been studied by drug development (drug under clinical investigation or already approved)

**F -** Drug target gene - disease pairs studied by GWAS where the disease has not been studied by drug development

clinical biomarker (e.g., LDL cholesterol for coronary heart disease) as well as directly with the disease endpoint. There may also be inconsistencies in the annotation of clinical endpoints to a coding system (e.g., non-small cell lung cancer and non-small cell lung carcinoma have different codes in the unified medical language system, UMLS). Nevertheless, with these caveats, we identified 1914 diseases covered by GWAS (17.2% of the total number of common human diseases) based on the mapping and manual curation of phenotype terms in the GWAS Catalog[43] and UK Biobank[38] to UMLS concepts (Supplementary Data 5).

Of the 1549 diseases with an approved treatment and/or a treatment that is or has been investigated in clinical or preclinical development, 703 had also been investigated by GWAS (Fig. 2, Supplementary Data 6), leaving 1211 diseases that have been the subject of investigation in GWAS, but which have yet to be investigated in drug development (Supplementary Data 5).

### Important subcategories of drug target-disease indication pairings

Based on the previous mappings, we generated sample spaces based on different sub-categories of drug target–disease indication pairings to help inform future genomic and drug development efforts.

**Sample space bounded by all protein-coding genes and diseases.** As a denominator, we generated a sample space bounded by 19,813 protein-coding genes annotated in Ensembl v.108 and 11,158 diseases, which produces ~221 million protein-disease indication pairings (221,073,454; labelled A in Fig. 3).

**Sample space bounded by the druggable genome and all human diseases.** Since not all proteins are readily targeted by small molecule drugs or monoclonal antibodies or peptide therapeutics, the sample space more relevant to drug development is bounded by 4729 genes encoding druggable targets[11] and the 11,158 human diseases, which produces ~52.8 million (52,766,182) drug target-disease indication pairings that might be the subject of drug development. This space is labelled B in Fig. 3.

**Sample space bounded by target-indication pairings under clinical investigation.** Having defined these key denominator values, we identified a sample space bounded by the indications and the targets that have

been the subject of clinical investigation but within which some, but not all target–indication pairings have been explored in drug development. This space, labelled 'C' in Fig. 3, is bounded by 1218 genes encoding the targets of drugs (Fig. 1) and 1401 diseases for which these targets are being or have been investigated in clinical phase drug development (Fig. 2), giving around 1.7 million (1,706,418) target-indication parings. Within this bounded space of ~1.7 million target indication pairings within sample space 'C', only 42,199 unique target-indication pairings (2.5%) have been explored in drug development. Sample space **C** represents only about 3.2% of the ~52.8 million drug target–indication pairings that could be studied (sample space **B**), and 0.8% of all ~221 million protein–disease pairings (sample space **A**).

**Sample space bounded by target-indication pairings for approved drugs.** We identified 755 targets of approved drugs (Fig. 1) for 612 disease indications (Fig. 2), giving a sample space (labelled **D** in Fig. 3) of 462,060 target indication pairs. Of these, the number of drug target-disease indication hypotheses that have been explored and led to approval within this bounded space is 5221 (1% of the maximum space, Supplementary Data 1). As for target-indication pairings investigated in clinical development, the coverage of targets and indications of approved drugs is uneven. Some diseases have many targets for approved drugs, for example, there are 37 genes encoding approved drug targets for the treatment of hypertension (27 ChEMBL target IDs, of which seven are the targets of adrenergic receptors blocking drugs), whereas other diseases (e.g., pituitary dwarfism) have treatments directed at a single target. The median number of drug targets per approved indication is two (first quartile: 1, second quartile: 4). Similarly, several drug targets have been approved for multiple indications, including different disease areas. For example, the glucocorticoid receptor is employed as a drug target for the treatment of around 111 diseases, including disorders of the blood, immune, circulatory, respiratory systems, and different cancers (Fig. 4, Supplementary Data 7).

**Diseases and targets evaluated both in GWAS and drug development.** We identified 755 targets of currently approved drugs (16% of all druggable targets) employed in the treatment of 612 diseases (5.5% of all 11,158 diseases). Of these diseases, 336 have also been studied in GWAS. It is through this intersection that it has been possible to show that GWAS

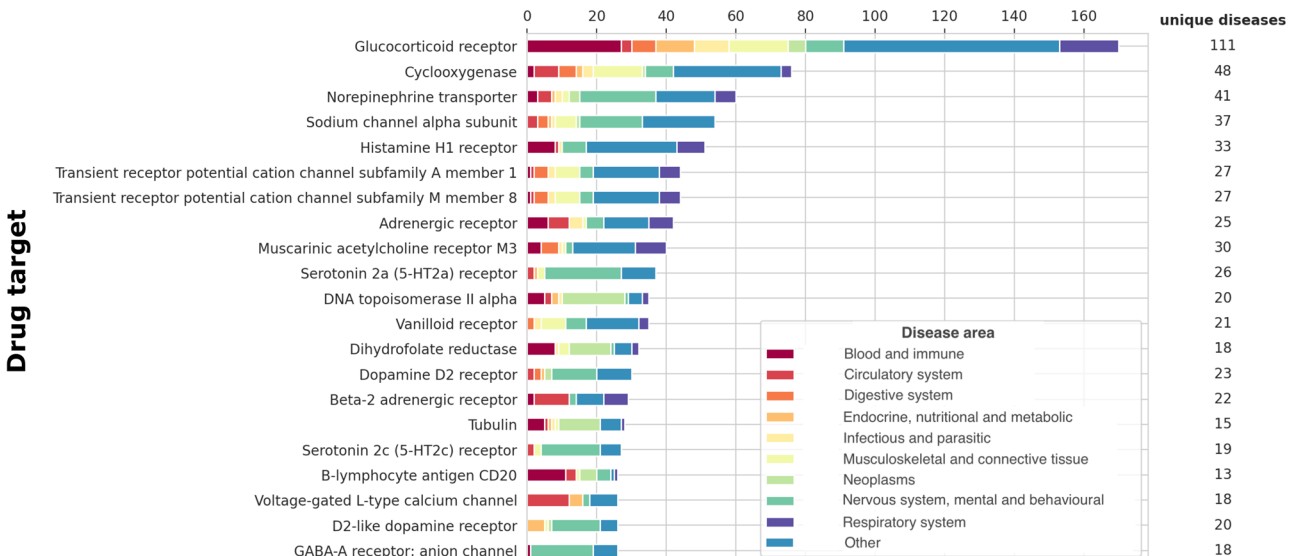

**Fig. 4 | Disease indications with an approved treatment by drug target.** Number of disease indications by drug target and disease area, with the number of unique diseases on the right, as one disease could be classified in multiple disease categories (e.g., multiple myeloma in MeSH is classified as a disease of the circulatory system, blood and immune and neoplasm). Only drug targets with more than 25 indications with an approved treatment are shown.

have frequently rediscovered established drug targets for the corresponding diseases[11–13]. The 1234 targets of drugs that are or have been the subject of clinical investigation (which includes the targets of approved drugs) have been or are being evaluated for the treatment of 1549 diseases.

Prior research has shown that drugs for which the target-indication pairing has genetic support have higher rates of approval[14–18]. However, of the 1401 disease indications being evaluated in clinical development, only 661 have previously been the subject of a GWAS. Of these 1401 disease indications, 476 have already approved treatment, and of the remaining 925 diseases, 363 have been studied by a GWAS.

## Discussion

Previous research has shown that human genetic evidence could support drug development[11,14,15,31]. However, the extent to which genomic efforts, specifically GWAS, align with ongoing drug development efforts and unmet needs has not been explored in detail. The current analysis shows: (1) Only a small fraction of the 11,158 diseases curated in the human DO have been investigated in drug development (14%; 1549 out of 11,158 diseases) or GWAS (17%; 1914 out of 11,158 diseases); (2) Of diseases being pursued in clinical phase drug development, only 47% have been the subject of a GWAS (661 out of 1401); and (3) Even for the 661 diseases that are the subject of ongoing clinical phase drug development and have been covered by GWAS, it remains uncertain how many of the specific target-indication pairings have genetic support. The construction of a sample space of disease and targets, including subsets of target-disease pairings that have been covered by GWAS (which interrogates all possible targets by design) and clinical phase drug development, can help generate insights into how these complementary efforts can be utilised in concert.

The results presented in this analysis represent the first systematic survey of the coverage, overlap and divergence of human genetic studies and diseases investigated in pharmaceutical research and development. One of the strengths of this analysis is that the data used were available in the public domain, which facilitates the revisiting of the estimates in the future. Another is that the analysis was stratified to show how the overlap between diseases with an approved treatment, a treatment that is or has been under clinical development and studied by GWAS, also differs at the level of individual disease. Standardisation of terms across data sources was challenging because of the different coding systems in the drug and GWAS databases and the lack of direct mapping across terminologies. By using the UMLS as an anchoring ontology to standardise the diseases across data sources and including a step of manual curation of the disease terms and areas, the error due to inaccurate mapping cross-databases was reduced.

The intersection between targets of approved drugs and diseases studied by GWAS can help identify new indications for existing approved drugs (Fig. 5). On the other hand, the intersection between targets of drugs that are or have been the subject of clinical investigation and diseases studied by GWAS can lead to potential repurposing opportunities of drugs that proved safe but lacked efficacy for their intended indication, or for indication expansion of approved drugs (Fig. 5). Both indication expansion and repurposing are attractive alternatives to de novo drug development, mainly because such compounds have been proven to engage well-characterised targets and the medicines have proven safe in clinical trials, which leads to a reduction in costs and development timelines[44]. High failure rates in clinical phase drug development have heightened interest in the therapeutic repurposing of drugs that failed in their originally intended indication for lack of efficacy. Previous modelling studies have suggested that any given drug target might be useful in the treatment of multiple diseases[31]. There are well-established examples of this. Beta-adrenoceptor antagonists are used in the treatment of hypertension, coronary heart disease, heart failure, portal hypertension, migraine, anxiety, tremors in thyrotoxicosis and infantile haemangiomas. SGLT2 inhibitors developed for diabetes have now been approved for use in heart failure (reduced and preserved ejection fraction) and in chronic kidney disease. GWAS can be used as a source of evidence for drug target identification. One route to expanding the indications of licensed drugs or those in development or to repurpose investigational drugs that fail in their intended indication, would be to systematically interrogate the association of variants in the genes encoding the targets of these drugs in GWAS data. Since GWAS have already investigated 1914 diseases, there is already a large dataset that could be utilised for this purpose. For example, the interleukin-6 receptor is the target of an approved drug (tocilizumab) used in the treatment of rheumatoid arthritis. Encouragingly, the gene encoding this receptor has also been identified using GWAS of coronary heart disease, abdominal aortic aneurysm and atrial fibrillation, suggesting a number of indication expansion opportunities[21,45,46]. Another example is the interleukin-23 receptor inhibitor ustekinumab, which was originally

## EXPLOITING EXISTING OPPORTUNITIES FOR DRUG TARGET IDENTIFICATION

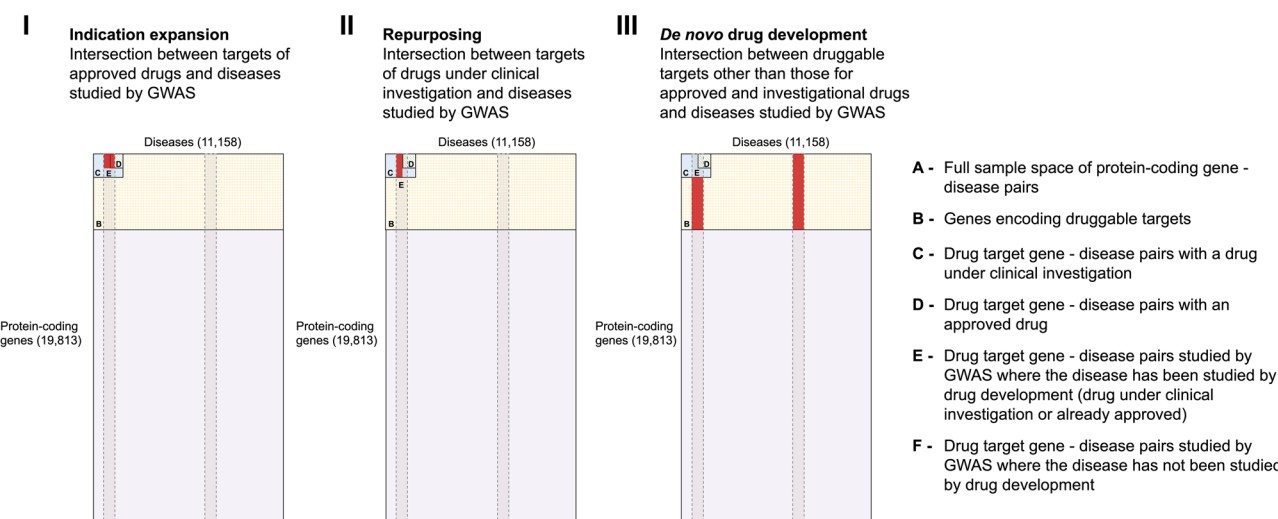

**Fig. 5 | Exploiting existing opportunities for drug target identification.** In each subfigure, the complete sample set (**A**) is bounded by the total number of protein-coding genes (19,995) and the sum total of common, complex human diseases (11,158). The subset of all potentially druggable target-disease indication pairings is indicated by subset **B**, the drug target-disease indication pairings studied in clinical phase drug development by subset **C**, and the target-disease indication pairings of approved drugs by subset **D**. The vertical lines represent diseases studied by GWAS on the assumption that GWAS interrogate all genes in the human genome (subset **E** and **F**). The presence of two GWAS subsets illustrates the point that only a subset of diseases studied in GWAS have also been the subject of drug development (**E**). The red area in subfigure I indicates de intersection between targets of approved drugs and diseases studied by GWAS, where there could be potential for indication expansion informed by GWAS. The red area in subfigure II indicates de intersection between targets of drugs under clinical investigation and diseases studied by GWAS, where there could be potential for repurposing informed by GWAS. The red area in subfigure III indicates the intersection between targets of druggable targets other than those for approved and investigational drugs and diseases studied by GWAS, where there could be potential for de novo drug development informed by GWAS.

intended to treat psoriasis and, after identifying a GWAS signal for Crohn's disease was investigated for such indication and eventually approved in 2017[47–49]. In fact, in a recent analysis, Trajanoska et al., used data from the Open Targets Platform to identify genetic evidence for approved drugs, considering that a minimum of five years separates an original genetic observation from the approval of a derived and found that 47 first-in-class therapies for 40 targets had been genetically driven[50].

In addition, the sample space of human targets and diseases could also inform de novo drug development for druggable targets and disease indication pairings that have yet to be investigated. One way would be by increasing the range of druggable targets (space **B** in Fig. 4). This is becoming possible through technological developments. These include (1) the growing use of monoclonal antibodies and the development of cyclic peptides as therapeutics for protein targets that lack a binding pocket amenable to targeting by conventional small molecule therapeutics[47,49,51–53], and (2) the targeting of RNAs rather than proteins using RNA silencing approaches and the emergence of CRISPR-Case 9-based gene editing in cases for proteins that remain difficult to drug[54–56]. A complementary approach necessary to map the expanded range of druggable targets to the correct diseases is to increase the range of diseases that have been studied in GWAS. This is becoming possible by the greater deployment of genetic studies within large national biobanks linked to healthcare data[27,57–59], and even in healthcare systems[60–62].

There are groups of targets that could especially benefit from having genetic support. For example, identifying soluble or secreted protein targets with genetic evidence for a particular disease represents an attractive venture since such proteins are readily targeted by monoclonal antibodies or peptides, which typically exhibit higher selectivity and reduced development timelines compared to small molecules[63]. Information on the set of human secreted proteins (the 'secretome'[64]) is available in the public domain, and researchers and the pharmaceutical industry could use these resources to identify high priority putative circulating protein targets. In addition to therapeutics that exert their action at the protein level, novel therapies based on RNA silencing or interference provide a solution to downregulate protein targets that are resistant to small or large molecule therapeutics[54]. While this technique is currently limited by the effective delivery of the RNA into the target tissue, existing technologies support efficient targeting of the liver with RNA-based therapeutics[65]. Therefore, genetically supported targets with elevated gene expression in the liver may be prioritised for RNA silencing therapy.

Furthermore, the sample space of human protein targets and diseases can be used to inform new drug development programs and research (Fig. 6). For example, only 17% of human diseases have been investigated in a GWAS, and over 8000 diseases exist without an approved treatment or under clinical investigation. Prioritising diseases for genomic analysis with a view to generating critical evidence for drug development is one of the numerous applications of the current analysis. Large biobanks with genetic data linked to routinely collected primary and secondary care health records provide an opportunity to investigate targets with genetic support in conditions with unmet medical needs or to increase the power in diseases where a GWAS is available, but the number of cases was not sufficient to reliably identify genetic associations.

Some limitations are worthy of note. Our analysis used the number of disease terms in the DO to estimate the sample space bounded by all protein-coding genes and diseases. DO is updated regularly and includes rare, common and complex diseases, however, it may be incomplete and not capture all medical conditions. Nonetheless, our analysis mapped disease indications studied by GWAS to disease indications investigated in drug development to estimate the extent GWAS and pharmaceutical research and development efforts overlap, and thus, DO was simply used to provide a denominator for the total number of diseases. We could have alternatively used ICD-10 or HPO (Table 1) to perform the calculation, which would not have impacted the findings, as ICD-10 and HPO include 12,318 and 16,601, respectively, a very similar number to the diseases included in DO (11,158). Our analysis focused on common polygenic human diseases, which are the ones subjected to GWAS. Separately, rare loss-of-function variants causing

## CREATING NEW OPPORTUNITIES FOR DRUG TARGET IDENTIFICATION

**I** New disease indications with genetic support

- Expanding the scope of GWAS
- GWAS in routine healthcare systems (e.g., EHR datasets)

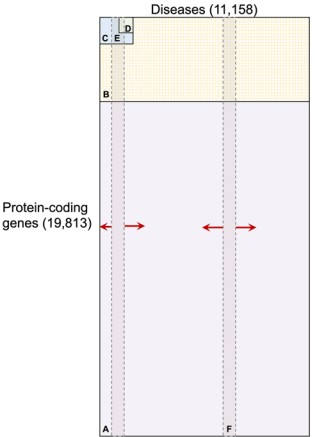

**II** Increasing the universe of therapeutic targets

- Protein targets
- mRNA targets (liver/tissue constraints)
- Gene editing (liver/tissue constraints)

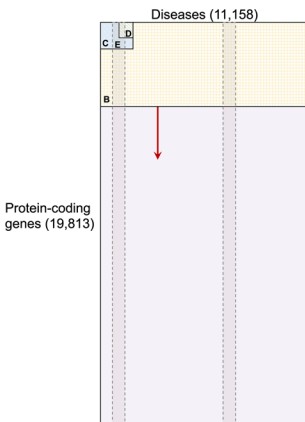

**A -** Full sample space of protein-coding gene - disease pairs

**B -** Genes encoding druggable targets

**C -** Drug target gene - disease pairs with a drug under clinical investigation

**D -** Drug target gene - disease pairs with an approved drug

**E -** Drug target gene - disease pairs studied by GWAS where the disease has been studied by drug development (drug under clinical investigation or already approved)

**F -** Drug target gene - disease pairs studied by GWAS where the disease has not been studied by drug development

**Fig. 6 | Creating new opportunities for drug target identification.** In each subfigure, the complete sample set (**A**) is bounded by the total number of protein-coding genes (19,813) and the sum total of common, complex human diseases (11,158). The subset of all potentially druggable target-disease indication pairings is indicated by subset **B**, the drug target-disease indication pairings studied in clinical phase drug development by subset **C**, and the target-disease indication pairings of approved drugs by subset **D**. The vertical lines represent diseases studied by GWAS on the assumption that GWAS interrogate all genes in the human genome (subset **E** and **F**). The presence of two GWAS subsets illustrates the point that only a subset of diseases studied in GWAS have also been the subject of drug development (**E**). The red arrows in Subfigure I illustrate the expansion of the subsets E and F by expanding the scope of GWAS and data sources. The red arrow in Subfigure II illustrates the expansion of subset **B** by expanding the genes encoding druggable targets.

monogenic disorders have correctly predicted the safety and phenotypic effect of pharmacological inhibition, but these were outside the scope of the present analysis[66]. We also relied on the trait and indication mappings provided by the GWAS Catalog and ChEMBL, which may not include the necessary granularity. For example, for some compounds, ChEMBL includes general indications such as digestive system diseases or cardiovascular diseases. We used the UMLS to map across databases, however, further efforts are needed to harmonise the disease ontologies used in GWAS and drug development. Importantly, we sourced GWAS data from the public domain, however, all the GWAS that may have been performed, including additional diseases to those in the GWAS Catalog, may not have been disseminated to the public which would underestimate the number of existing diseases studied in GWAS.

Notably, the potential of GWAS for drug development relies on assigning genetic associations from GWAS data to a causal gene, which remains a challenge because association signals from variants in high linkage disequilibrium (LD) may span multiple genes. Several gold-standard datasets have been used to explore the best approach to assign GWAS signals to genes. These 'truth' sets include genes whose perturbation causes a Mendelian form of a common disease[12], the set of expression, protein and metabolite QTLs[67,68], manually curated target examples from the literature[69], and approved drug target-indication pairings where the indication has been rediscovered by GWAS[11,69]. Numerous statistical and computational approaches have been suggested to assign GWAS signals to genes, including co-localisation[70], and machine-learning techniques[69]. Yet, physical proximity remains the simplest and most widely used proxy to map association signals to genes[68,71]. Although examples exist where the closest gene is not the likely causal gene[72,73], several studies using a set of genes with well-validated causal relationships to disease have revealed the closest gene to a GWAS signal to be the causal gene in about two-thirds of cases[68], and have shown that the relative distance to the gene is the best single predictor of a causal gene[69].

Our approach was based on the current 'druggable genome' (the set of genes encoding proteins that are or may be readily drugged by small or large molecule drugs). However, this concept is an evolving entity. Whenever a new protein is drugged, this opens the possibility of drugging structurally similar proteins, thereby expanding the bounds of the druggable genome. Where proteins remain difficult to drug, RNA silencing or gene editing now offer alternative therapeutic approaches[74–76]. Our analysis was also based on semantic mapping, but several commercial and academic efforts are employing artificial intelligence for target identification and drug discovery[77,78]. The application of artificial intelligence, and computer modelling have predicted protein structures and revealed previously unknown protein motifs potentially turning undruggable protein targets into druggable ones[79]. The available information on drugs in development may be incomplete or inaccessible for commercial reasons, which may lead to an underestimation of the number of diseases studied in drug development, particularly for the preclinical candidates which did not progress to clinical trials. Regarding the number of diseases investigated by GWAS, some diseases could have been studied through a validated clinical biomarker which may not have been well captured by the approach we used. In addition, this analysis did not consider disease prevalence which is an important factor for the design, interpretability, and future direction of GWAS, as well as for the evaluation of diseases with unmet clinical needs. Equally, mortality or morbidity are also important variables to consider when contextualising the findings and defining research and pharmaceutical strategies. Many diseases are not tractable by therapeutic agents (i.e., congenital malformations) and require surgical or device-based treatments. For those diseases, GWAS can provide insights into the molecular basis of the disease, but genetic associations with causal genes may not be used to inform drug development. In other cases, key targetable mechanisms (e.g., autoimmunity) may trigger the development of a disease but may no longer be usefully targeted by the time the disease is manifest (e.g. type 1 diabetes). Finally, not all human diseases may need to be treated with a therapeutic

intervention, particularly if the medicinal product has a risk of harm and may instead be best prevented through public health interventions.

In conclusion, we have systematically mapped drug development and genomic discovery efforts in common diseases to produce target-disease indication lists. These lists could help the pharmaceutical industry to prioritize future drug development efforts based on genetic evidence, academia to prioritize future GWAS for diseases without effective treatments, and both sectors to harness genetic evidence to expand the indications for licensed drugs or to identify repurposing opportunities for clinical candidates that failed in their originally intended indication.

## Data availability

The Human Disease Ontology (DO)[30] is publicly available and can be accessed at https://github.com/DiseaseOntology/HumanDiseaseOntology/tree/master/src/ontology. Data on approved drugs and compounds under development is publicly available at ChEMBL v31[35] and can be accessed at https://chembl.gitbook.io/chembl-interface-documentation/downloads. All source GWAS data used throughout the paper are publicly available and were obtained from the GWAS Catalog v1.0.3[37] (https://www.ebi.ac.uk/gwas/downloads) and from Neale data, UK biobank GWAS Round 2, Results shared 1st August 2018[38] (https://www.nealelab.is/uk-biobank). Supplementary Data 1 lists all human genes encoding druggable targets and the maximum clinical phase reached by indication. Supplementary Data 2 includes diseases that are the indication of approved drugs. Supplementary Data 3 includes diseases that are or have been the indication of drugs in clinical development. Supplementary Data 4 includes diseases that are or have been indications for drugs in preclinical development. Supplementary Data 5 includes diseases subjected to GWAS and deposited in the GWAS Catalog or studied by Neale lab. Supplementary Data 6 shows diseases subjected to GWAS and deposited in the GWAS Catalog or studied by Neale lab and subjected to drug development (preclinical, clinical or approved). Supplementary Data 7 includes compounds and their targets for drug targets with more than 25 indications with an approved treatment. The source data for Fig. 1 is in Supplementary Data 1. The source data for Fig. 1 is in Supplementary Data 2–6. The number of diseases in Figs. 3, 5 and 6 was sourced from the Human Disease Ontology (DO)[30] (https://github.com/DiseaseOntology/HumanDiseaseOntology/tree/master/src/ontology) on 29 September 2022, and the total number of protein-coding genes in the human genome from Ensembl v.108 (https://ftp.ensembl.org/pub/). The source data for Fig. 4 is in Supplementary Data 7.

## Code availability

The code to perform the analyses and underlying each figure has been deposited in the UCL Research Data Repository under accession code https://doi.org/10.5522/04/25541392.v1[80].

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

## Acknowledgements

M.G.M. was supported by a BHF Fellowship FS/17/70/33482. A.F.S. was supported by BHF grant PG/22/10989, and the UCL BHF Research Accelerator AA/18/6/34223. A.N.W. was funded by a NIHR Academic Clinical Lectureship and a Wellcome Trust Collaborative Award (224390/Z/21/Z). C.T. was supported by a UCL UKRI Centre for Doctoral Training in AI-enabled Healthcare studentship (EP/S021612/1), MRC Clinical Top-Up, and a studentship from the NIHR Biomedical Research Centre at University College London Hospital NHS Trust. J.E. was supported by the UKRI/NIHR-funded Multimorbidity Mechanism and Therapeutics Research Collaborative (MR/V033867/1). A.T. was supported by the NIHR-funded CONVALESCENCE programme (COV-LT-0009). R.M. was funded by a NIHR Academic Clinical Fellowship. R.S. was supported by the UKRI/NIHR-funded Multimorbidity Mechanism and Therapeutics Research Collaborative (MR/V033867/1) and HDR UK. A.S. was supported by NIHR (AI_AWARD01864 and COV-LT-0009), Innovate UK (Horizon Europe Guarantee for DataTools4Heart) and EPSRC (EP/Y018087/1). S.D. is supported by (1) the UCL British Heart Foundation Accelerator (AA/18/6/34223), (2) the UCL NIHR Biomedical Research Centre (NIHR203328), (3) the UKRI/NIHR funded Multimorbidity Mechanism and Therapeutics Research Collaborative (MR/V033867/1), (4) Health Data Research UK Phenomics and Prognostic Atlas Theme, (5) the BHF Data Science Centre led by HDR UK (grant number SP/19/3/34678), (6) the CVD-COVID-UK/COVID-IMPACT Consortium, (7) the Longitudinal Health and Wellbeing strand of the National Core Studies programme (MC_PC_20030: MC_PC_20059), and (8) The NIHR funded CONVALESCENCE programme (COV-LT-0009). M.P. is supported by the UKRI/NIHR-funded Multimorbidity Mechanism and Therapeutics Research Collaborative (MR/V033867/1) and HDR UK. H.H. is supported by HDR UK and the NIHR Biomedical Research Centre, University College London Hospitals NHS Trust. A.H. was supported by the UCL British Heart Foundation Accelerator (AA/18/6/34223), the UCL NIHR Biomedical Research Centre (NIHR203328), and the UKRI/NIHR funded Multimorbidity Mechanism and Therapeutics Research Collaborative (MR/V033867/1), and an NIHR Senior Investigator Award NIHR202383. C.F. was supported by the UCL British Heart Foundation Accelerator (AA/18/6/34223), the UCL NIHR Biomedical Research Centre (NIHR203328), the UKRI/NIHR funded Multimorbidity Mechanism and Therapeutics Research Collaborative (MR/V033867/1), and an NIHR Senior Investigator Award NIHR202383. The authors are grateful to the studies and consortia that provided summary association results. A.F.S. and C.F. have received unrestricted funding from New Amsterdam Pharma. C.T., C.Y., A.T., and S.D. have received funding paid to University College London from GlaxoSmithKline (GSK), outside of the scope of the submitted work. M.P. has received partnership funding for the following: MRC Clinical Pharmacology Training Scheme (co-funded by MRC and Roche, UCB, Eli Lilly, and Novartis). M.P. has developed an HLA genotyping panel with MC Diagnostics but does not benefit financially from this. He is part of the IMI Consortium ARDAT. A.D.H. and C.F. are co-investigators on a grant from Pfizer to identify potential therapeutic targets for heart failure using human genomics. A.F.S. is an Editorial Board Member for Communications Medicine but was not involved in the editorial review or peer review, nor in the decision to publish this article. The views expressed in this study are the personal views of M.G.M. and do not represent the views of her current employer, the Spanish Agency for Medicines and Medical Devices.

## Author contributions

M.G.M., A.D.H., and C.F. contributed to the idea and design of the study. M.G.M. performed the analyses. M.G.M. and A.D.H. contributed to the first draft of the paper. A.F.S, A.W., C.T., C.Y., J.E., A.T., R.M., R.S., C.L., A.S., S.D., M.P., H.H., and C.F. contributed to and approved the final version of the paper.

## Competing interests

A.W., J.E., R.M., R.S., C.L., A.D.S., and H.H. declare no competing interests.
