## [Peer Review File · Communications Medicine]

This manuscript has been previously reviewed at another *Nature Portfolio* journal. This document only contains reviewer comments and rebuttal letters for versions considered at *Communications Medicine*.

REVIEWERS' COMMENTS:

Reviewer #1 (Remarks to the Author) - previous reviewer:

I appreciate all the clarifications in the revised manuscript, as I think it will help the reader better understand the claims. In this reviewer's opinion, a more explicit hypothesis and a better treatment of the disease/ontology space and gene assignment would have benefited this manuscript. Overall, it provides a complementary view to other analyses and a positive contribution to advancing the field. Thank you.

Reviewer #3 (Remarks to the Author) - new reviewer:

Thank you for the opportunity to review a revised version of the manuscript although I am not involved in the first round of review. I only have one minor comment where I would suggest to explain why certain databases are preferred over the others (For drugs in development). The arguments in the rebuttal letter could be of interest to readers.

Response to referees

Reviewer #1 (Remarks to the Author) - previous reviewer:

I appreciate all the clarifications in the revised manuscript, as I think it will help the reader better understand the claims. In this reviewer's opinion, a more explicit hypothesis and a better treatment of the disease/ontology space and gene assignment would have benefited this manuscript. Overall, it provides a complementary view to other analyses and a positive contribution to advancing the field. Thank you.

Response: We thank the reviewer for their positive comment.

Reviewer #3 (Remarks to the Author) - new reviewer:

Thank you for the opportunity to review a revised version of the manuscript although I am not involved in the first round of review. I only have one minor comment where I would suggest to explain why certain databases are preferred over the others (For drugs in development). The arguments in the rebuttal letter could be of interest to readers.

Response: We thank the reviewer for their comment. We have incorporated the arguments in the previous "Response to referees" letter regarding the selection of ChEMBL database over DrugBank, PharmaProjects or Patsnaps as follows:

Page 6

ChEMBL was selected as the data source for compound, target and indication information for the following reasons: i) it is publicly available, ii) it is manually curated, iii) it contains information on drugs that have been approved for treatment of a specific disease / diagnosis (an indication) within any region of the world.

Pharmaprojects³² or Patsnap³³ are private resources and thus, were not considered for this analysis. ChEMBL was selected over DrugBank as the latter only contains information on FDA-approved drugs³⁴.